# Multiple Parallel Fusion Network for Predicting Protein Subcellular Localization from Stimulated Raman Scattering (SRS) Microscopy Images in Living Cells

**DOI:** 10.3390/ijms231810827

**Published:** 2022-09-16

**Authors:** Zhihao Wei, Wu Liu, Weiyong Yu, Xi Liu, Ruiqing Yan, Qiang Liu, Qianjin Guo

**Affiliations:** 1Academy of Artificial Intelligence, Beijing Institute of Petrochemical Technology, Beijing 102617, China; 2School of Mechanical Engineering & Hydrogen Energy Research Centre, Beijing Institute of Petrochemical Technology, Beijing 102617, China

**Keywords:** label-free live cell imaging, protein subcellular localization, nonlinear optical microscopy, multiple parallel fusion network, deep learning

## Abstract

Stimulated Raman Scattering Microscopy (SRS) is a powerful tool for label-free detailed recognition and investigation of the cellular and subcellular structures of living cells. Determining subcellular protein localization from the cell level of SRS images is one of the basic goals of cell biology, which can not only provide useful clues for their functions and biological processes but also help to determine the priority and select the appropriate target for drug development. However, the bottleneck in predicting subcellular protein locations of SRS cell imaging lies in modeling complicated relationships concealed beneath the original cell imaging data owing to the spectral overlap information from different protein molecules. In this work, a multiple parallel fusion network, MPFnetwork, is proposed to study the subcellular locations from SRS images. This model used a multiple parallel fusion model to construct feature representations and combined multiple nonlinear decomposing algorithms as the automated subcellular detection method. Our experimental results showed that the MPFnetwork could achieve over 0.93 dice correlation between estimated and true fractions on SRS lung cancer cell datasets. In addition, we applied the MPFnetwork method to cell images for label-free prediction of several different subcellular components simultaneously, rather than using several fluorescent labels. These results open up a new method for the time-resolved study of subcellular components in different cells, especially cancer cells.

## 1. Introduction

Cells can be divided into different organelles for metabolic processes and complicated intra-cellular organizations. They are the basic biological, structural, and functional unit of all living organisms, and the dysfunction of organelles is often closely linked to the occurrence, development, and metastasis of tumors [1,2,3,4,5,6]. Accordingly, the detection and digging deeper into the structure, function, and micro-environment of cell organelles will help us further learn about the important role of organelles in life activities and provide effective suggestions for the diagnosis and treatment of cell organelle-related diseases [5,6,7,8,9,10]. However, the resolution of organelle structure in the natural tissue environment and its functional consequences are still not clear [6,7,8,9,10,11,12]. Accordingly, the capability of imaging, extracting, and exploring cells and their subcellular compartments is essential in various research fields, such as cell physiology and pathology, and is closely related to a variety of diseases.

Based on the above reasons, many related technologies have been developed and applied to cells and their subcellular compartment detection and research [13,14,15,16]. These detection methods mainly include traditional biological methods such as gel electrophoresis, protein immunoblotting, and mass spectrometry, while emerging microscopy technologies include electron microscopy, atomic force microscopy, and different type of optical imaging technologies such as fluorescence imaging technology, confocal microscopy imaging, phase contrast imaging, Raman imaging technology, and super-resolution fluorescence microscopy. These methods are extensively applied in single-cell investigations and offer an important way to study different angles of cell information [13,14,15,16,17,18,19,20,21,22].

Compared with other cell imaging methods, the optical method of single-cell imaging has certain improved properties, such as high detection sensitivity, high quality, and low cost, which tremendously boost the proceedings of non-destructive cell research [18,19,20,21,22]. Recently, large amounts of unlabeled optical imaging instruments have been utilized for cell surveys [18,19,20,21,22,23]. Compared with pathological images, which need staining, and fluorescent images, which need labeling, label-free optical imaging overcome the unfavorable influence of staining reagents on cytoactive and cell signal transduction and can be used for long time detection in tissues and living cells [20,21,22,23]. Therefore, there is increasing demand to develop advanced label-free cell optical imaging analysis methods to mine the rich information contained in the optical cell images [24,25,26].

Due to the successive enhancement and usability of sophisticated computing power and analytical methods in recent years, the deep neural network learning method has been prevalent in the field of label-free cell optical imaging techniques for deeply exploring cellular structure and morphological information [25,26,27,28]. In comparison with the conventional intelligence method, deep learning is able to automatically perform a series of target recognition, feature extraction, and analysis, which makes it possible to automatically discover image target features, and automatically explore feature levels and interactions [27,28,29,30]. The learning-enhanced cell optical image analysis model is capable of acquiring texture details from low-level source images and achieving higher resolution for label-free cell optical imaging techniques [29,30,31,32]. The deep learning pipeline of cell optical microscopy imaging can extract complex data representations in a hierarchical way, which is helpful for finding hidden cell structures from microscope images, such as the size of a single cell, the number of cells in a given area, the thickness of the cell wall, the spatial distribution between cells, subcellular components and their density, etc. [32,33,34,35,36,37,38,39,40,41,42,43,44,45,46]. The U-net model has been proven effective in semantic segmentation and label-free prediction for cell optical microscopy imaging [33]. However, for the multiple spectral channels, the spatial and spectral information are mixed during the feature encoding in a problematic fashion for image reconstruction in traditional Unet architecture. Although other recently reported modifications of Unet, such as UwU-Net architecture, are good at extracting local feature regions and can dedicate tunable free parameters to both spectral and spatial information, they still experience difficulty in capturing global representations [36].

Motivated by the above analysis, the goal of this work was to employ a multiple parallel fusion Deep Networks framework that boosts the performance of label-free cell optical imaging techniques when solving sub-cellular organelle location problems. In our work, we find that the multiple parallel fusion method (MPF)—which incorporates all the merit features of both high-resolution local detailed spatial information from CNN features and the global context information from transformers—presents a better way to predict the location of cellular organelles from label-free cell optical images compared with previous CNN-based self-attention methods. Moreover, we demonstrate that subcellular structures could be more precisely reconstructed with the combination of transformer and Unet than both methods working individually. The model also has a strong generalization ability and can be extended to the new cell imaging investigation.

## 2. Results

### 2.1. Experimental Settings

To compare the performance of different models, the setting of experimental parameters should be as consistent as possible. Firstly, the development, training, prediction, and image processing of all models were calculated by Pytorch platforms, and the graphics card of the server adopts Geforce RTX 3080. Secondly, during model training, the value of momentum was set at 0.9, the value of the batch size was set to 8, and the weight attenuation for the training neural network was set to 1 × 10^−4^. At the same time, the maximum number of epochs for the contrasting models was set at 200. In order to select the initial learning rate, a series of values was computed to test its training effect in the model. According to the experimental comparison, it was proved that 0.001 is the best choice to set as the initial learning rate.

The neural network training curves for three different prediction methods are shown in Figure 1. For the better-performing multiple parallel branch fusion model, as Figure 1 depicts, the training process only took about 120 steps until the training accuracy increased by over 96%. The error decay in Figure 1 demonstrates that the method with the MPFnet mechanism gained better performance on different training samples in comparison with the classical Unet and UwUnet model, where our strategy avoids overfitting because the error does not increase with the change of training mode, and the error attenuation remains stable.

### 2.2. Metrics for Performance Evaluation

In order to verify the credibility of predictions, five quantified metrics were applied to measure the performance of different prediction algorithms. All the evaluation metrics mentioned above can be consecutively calculated as follows.

The accuracy (*AY*) and overall accuracy (*OA*) are common standard metrics for predicting subcellular locations, which can be calculated as follows:(1)$$AY\left(i\right)=\frac{R\left(i\right)}{S\left(i\right)}$$
(2)$$OA=\frac{\sum_{i=1}^{10}R\left(i\right)}{\sum_{i=1}^{10}S\left(i\right)}$$
where *R*(*i*) is the correctly predicted values in the *i*th subcellular locations, and *S*(*i*) represents the total values in the *i*th subcellular locations.

Mean Intersection over Union (*MIo**U*) is another standard metric for segmentation purposes [47]. Intersection over Union (*IoU*) is a ratio computed on a per-class basis between the ground truth and the protein subcellular location prediction. Mean Intersection over Union is the average IoU ratio, which can be calculated as follows:(3)$$IoU=\frac{T⋂P}{T⋃P}$$
(4)$$MIoU=\frac{1}{k+1}\overset{k}{\underset{i=0}{\sum }}\frac{p_{ii}}{\sum_{j=0}^{k}p_{ij}+\sum_{j=0}^{k}p_{ji}-p_{ii}}$$
where it is assumed that the total of classes is (*k* + 1) and *p_ij_* is the number of pixels of class *i* inferred to class *j*. *p_ii_* represents the number of true positives, while *p_ij_* and *p_ji_* are usually interpreted as false positives and false negatives, respectively.

Pearson’s correlation coefficient (PCC) (*r_py_* ∈ [−1, 1]) is another metric that gives the relationships between the feature values and the predicted values by measuring the correlation between the pixels of the true and predicted images [48]. Given N sample pairs {(*p*_1_, *y*_1_), …, (*p*_N_, *y*_N_)}, we obtain:(5)$$r_{py}=\frac{\sum_{i=1}^{N}(p_{i}-\bar{p})\left(y_{i}-\bar{y}\right)}{\sqrt{\left(\left(\sum_{i=1}^{N}{\left(p_{i}-\bar{p}\right)}^{2}\right)\left(\sum_{i=1}^{N}{\left(y_{i}-\bar{y}\right)}^{2}\right)\right)}}$$
where $\bar{p}$ and $\bar{y}$ are the sample means. Note that when *p_i_* and *y_i_* are binary, *r_py_* becomes the Matthews correlation coefficient, which is known to be more informative than the F_1_ score (Dice coefficient) on imbalanced datasets.

*MSE* (Mean Square Error) is a function that is used to evaluate the difference between the targeted values and the predicted values [49]. *RMSE* (Root Mean Square Error) further evaluates the spatial detail information between images, while *NRMSE* (Normalized Root Mean Square Error) normalizes *RMSE* for easier observation and comparison. For the image prediction work, the *NRMS* can be applied in computing the accuracy between a pixel in the predicted image and the same pixel in the true image, which is obtained by:(6)$$MSE=\frac{1}{M\times N}\overset{M}{\underset{i=1}{\sum }}\overset{N}{\underset{j=1}{\sum }}{\left(u^{\prime }\left(i,j\right)-u\left(i,j\right)\right)}^{2}$$
(7)$$RMSE=\sqrt{MSE\left(u^{\prime },u\right)}$$
(8)$$NRMSE=\frac{\sqrt{\frac{1}{M\;\times \;N}\sum_{i=1}^{M}\sum_{j=1}^{N}{\left(u^{\prime }\left(i,j\right)-u\left(i,j\right)\right)}^{2}}}{u^{\prime }{\left(i,j\right)}_{max}-u^{\prime }{\left(i,j\right)}_{min}}$$
where $u^{\prime }\left(i,j\right),\;u\left(i,j\right)\;$ represent the image to be evaluated and the original image, respectively. *N* represents the length and width of the image.

Peak Signal to Noise Ratio (*PSNR*) is the most commonly used metric in image quality assessment, which can be obtained by:(9)$$PSNR=10\;{log}_{10}\left(\frac{m_{x}\times m_{y}\times V_{max}^{2}}{\sum_{r,t}{\left[t\left(x,y\right)-d\left(x,y\right)\right]}^{2}}\right)$$
where *V_max_* denotes the maximum predicted value of the source image. *t(x*,*y)* is the matrix of the raw source image, *d(x*,*y)* is the matrix of the noise-removed image. (*x*,*y*) denotes the pixel coordinate in a given *m_x_* × *m_y_* image.

Structural similarity index (SSIM) can be used as a quality evaluation index for similarity comparison among image prediction results, which can be obtained by:(10)$$s\left(x,y\right)=\frac{\sigma_{xy}+c_{3}}{\sigma_{x}\sigma_{y}+c_{3}}$$
(11)$$l\left(x,y\right)=\frac{2\mu_{x}\mu_{y}+c_{1}}{\mu_{x}^{2}+\mu_{y}^{2}+c_{1}}$$
(12)$$c\left(x,y\right)=\frac{2\sigma_{x}\sigma_{y}+c_{2}}{\sigma_{x}^{2}+\sigma_{y}^{2}+c_{2}}$$

The SSIM value is calculated for signals and images after combining Equations (10)–(12) as:(13)$$SSIM\left(x,y\right)={\left[l\left(x,y\right)\right]}^{m}{\left[c\left(x,y\right)\right]}^{n}{\left[s\left(x,y\right)\right]}^{p}\;\;\;\;\;\;\;\;\;\;\;\;\;\;\;\;\;\;\;\;\;=\frac{\left(2\mu_{x}\mu_{y}+C_{1}\right)\left(2\sigma_{xy}+C_{2}\right)}{\left(\mu_{x}^{2}+\mu_{y}^{2}+C_{1}\right)\left(\sigma_{x}^{2}+\sigma_{y}^{2}+C_{2}\right)}$$
where *m*, *n*, and *p* denote the magnitude values of the structure component *s(x*,*y)*, the luminance component *l(x*,*y)*, and the contrast component *c(x*,*y)*, respectively*. µ_x_*, *µ_y_* is the average of *x_i_*, *y_i_*. *σ_x_*, *σ_y_* is the variance of *x_i_*, *y_i_*.

The Dice Similarity Coefficient (DSCs), also named Sørensen–Dice similarity, is another one of the frequently used metrics in medical image competitions. It is a collective similarity metric, which is usually used to calculate the similarity of two segmented images. The calculation formula of the Dice coefficient is as follows:(14)$$Dice=\frac{2\times \left(T⋂P\right)}{T⋃P}$$
where the *Dice* similarity coefficient value threshold is [0, 1]. The best result of prediction or segmentation in a medical image is 1, and the worst result is 0.

### 2.3. The Analysis of the Performance of Multiple Parallel Fusion Deep Networks

The experiment of this section mainly discusses the results of the labeling-free method based on deep learning for protein subcellular localization from femtosecond stimulated Raman spectroscopic microscope images. Compared with other classical optical imaging methods, Stimulated Raman spectroscopy imaging has the advantages of not requiring fluorescent molecular markers and obtaining more information. However, the rich and overlapped information in the same collected image also causes difficulties for image analysis and feature extraction. Although some of the label-free staining methods based on Raman imaging have shown promising results in some organelles, there is still a lack of rich and effective means to predict the subtle changes of Raman spectra for single organelles.

The results for subcellular localizations can be seen in Figure 2A,B. The output results for different organelle locations, including nuclei (second column), mitochondria (third column), and endoplasmic reticulum (right) from the single raw SRS imaging cell (left), are shown in Figure 2A. One SRS raw image for A549 lung cancer cells from ATCC was output at the different positions of the multiple fusion model at the same time to determine the accuracy of subcellular localization predictions, including nuclei (top row), mitochondria (third column), and endoplasmic reticulum (bottom row). Figure 2B shows the output of nuclei (top row), mitochondria (third column), and endoplasmic reticulum (bottom row) from the transformer branch in the first column. The output from the bifusion branch and multi-model are shown in the second and third columns, respectively.

### 2.4. Comparison of the Performance with Various Prediction Models

In this section, predicting experiments of subcellular organelle localization results are investigated and compared among different deep learning models from label-free SRS microscopy images. Even though the traditional imaging-based pipeline has cells stained, SRS imaging can give more information on cell shape and subcellular structure without using molecular probes. On the other hand, it also produces some disadvantages, such as low-contrast and complex images, which create obstacles to clearly indicating the biochemical features of cells. Therefore, for SRS cell imaging, some challenges still exist in using these deep learning-based methods to clearly identify, segment, and quantify each subcellular structure in cell optical images. As a result, some advanced analysis methods are needed to explore the rich information hidden in cell images. Based on the above reasons, the new MPFnet method is proposed in this work, which bridges the transform model and convolutional neural networks to automatically segment organelles.

To demonstrate the application of deep learning models in label-free organelle prediction, we used fluorescence imaging of the fixed lung cancer cells as a ground-truth model and SRS microscopy images as the source image model. In Figure 3, the first column shows a live-cell Raman optical image, the second column is ground-truth fluorescence images taken after the cells are stained, and the following three columns are predicted fluorescence cell images with the UwUnet method, Unet method, and MPFnet method, respectively. From the experimental analysis results, we can see that the MPFnet method can accurately predict the location of each organelle from cell optical imaging data at the same time.

In order to quantitatively compare and analyze the effects of different prediction methods, we calculated the quantitative metrics: NRMSE, SSIM, PCC, DICE, and mean IoU to explore the differences between predicted and expected results of different methods, to compare the prediction performance between the methods proposed in this work with other classical methods. We firstly give a comparison of the prediction performance with the mean pixel accuracy curves and radar chart for five quantitative metrics, including NRMSE, SSIM, PCC, DICE, and mean IoU among Unet, UwUnet, and MPFnet models, as shown in Figure 4. One can observe from Figure 4 that the MPFnet ensemble method achieved the highest mean pixel accuracy of 0.92.

To verify the valuable quantitative evaluation parameter in subcellular organelle localization, the segmentation performance of all trained networks was also assessed by the Dice similarity coefficients to evaluate the similarity between the ground truth fluorescence labels and model-generated images (Figure 5). Compared to the observed datasets, the Dice similarity of the prediction for the nuclei, endoplasmic reticulum, and mitochondria task set was the highest for MPFnet and the lowest for the Unet approach.

The lowest Dice similarity coefficient of the Unet method for the nuclei prediction task set was attributed to the edges of the bony structures not being included in the gold standard as it lacked the long-distance segmenting in these whole regions. The lowest Dice similarity coefficients of the Unet method for the mitochondria and endoplasmic reticulum prediction task sets were due to the larger excluded region of the outer contour area in its gold standard, while this excluded region was segmented by the Unet method.

In addition, we also measured the accuracy of label-free prediction algorithms using the mean Intersection Over Union (IOU) evaluation metric. Here, the mIOU for each epoch comparison among different methods is shown in Figure 6a. Then, we used the box plot graph to give a more visual and intuitive representation of the quantitative evaluation of mIOU parameters on the prediction results of different algorithms (Figure 6). The box plot in Figure 6 shows five statistics in the data: minimum, first quartile, median, third quartile, and maximum. In Figure 6, the minimum value is represented by the extension of the red lines at the bottom, while the maximum value is represented by the extension of the red line at the top. The range of these two black lines refers to the mIOU accuracy range, and the top and bottom of the box refer to the accuracy of the upper quartile (=0.75) and lower quartile (=0.25), respectively. The blue solid line in the box indicates the median accuracy. It can be seen from Figure 6b–e that compared with other methods, the MPFnet method achieves the best performance on all the nuclei, mitochondria, and endoplasmic reticulum datasets. Compared with the observed datasets, MPFnet performed favorably by metric mIOU with 0.879, 0.876, and 0.881 for the nuclei, mitochondria, and endoplasmic reticulum task set, respectively, against alternative UwUnet approaches with mIOU mean values of 0.852, 0.861, 0.854. The classical Unet approach performed significantly worse with mIOU mean values of 0.716, 0.771, and 0.731.

To further characterize the predictive performance of three variants of the deep learning-based predictor on organelle (nuclei, mitochondria, and endoplasmic reticulum) segmentation tasks and to give comparable measures, we also provide Pearson’s correlation coefficient (PCC) to quantify the accuracy of the predictions. The PCC similarity value is used here to determine the similarity between the pixels of the truth and predicted images with various deep learning methods.

The top left radar chart of the PCC in Figure 7a shows the PCC metric value with three different test models for the cell SRS images. The box plot graph in Figure 7b–e provides a more visual and intuitive representation of the quantitative evaluation of PCC parameters on the prediction results of different algorithms. Compared to the observed datasets, MPFnet performed favorably in PCC mean value for the nuclei, endoplasmic reticulum, and mitochondria task sets, respectively, against the alternative UwUnet and classical Unet approaches. Comparing the PCC value with other methods, MPFnet results had good advantages over other methods. Therefore, the PCC similarity value in Figure 7a–e shows that the MPFnet method achieved top performance on all the nuclei, mitochondria, and endoplasmic reticulum datasets. Based on the PCC coefficients, our selected MPFnet method scheme achieved the best performance in all data sets.

Moreover, the NRMSE metric value was also employed in this section to assess the effectiveness of the different models by comparing the predicted images and the actual images. The top-right side of Figure 8a shows the radar chart of the NRMSE with three different test models for the cell images. Compared with other methods, it can be seen in Figure 8a–e that the MPFnet method achieved the top performance on all nuclei, endoplasmic reticulum, and mitochondria datasets. Compared to the observed datasets, MPFnet performed favorably in NRMSE mean values of the nuclei, endoplasmic reticulum, and mitochondria task sets, respectively, against the alternative UwUnet and classical Unet approaches.

At last, the SSIM index, which is based on structure, luminance, and contrast comparison, was applied for quality assessment in various different prediction methods. The radar chart of the SSIM quality assurance with three different test models for the cell images is shown in Figure 9a. A small SSIM index appeared in all data sets in the Unet method corresponding to the target area of prediction, having a relatively large difference between the ground truth fluorescence labels and model-generated images. Moreover, from the box plot of SSIM values in the all-organelles (nuclei, mitochondria, and endoplasmic reticulum) prediction task set with three different methods, we can obtain more definite quantitative comparison information. As shown in Figure 9b–e, MPFnet performed favorably in SSIM mean value for the nuclei, endoplasmic reticulum, and mitochondria task sets against alternative UwUnet and classical Unet approaches. The MPFnet method achieved the top SSIM performance on all nuclei, endoplasmic reticulum, and mitochondria datasets.

In terms of explicit and quantitative analysis, the detailed evaluation results and calculations are shown in Table 1, Table 2 and Table 3. For different prediction models, five different quantitative parameters (NRMSE, SSIM, PCC, Dice, and mIOU) were computed to compare and analyze the accuracy of protein subcellular localization from label-free live cell imaging. Table 1, Table 2 and Table 3 present the label-free prediction results of three variants of deep learning-based predictors on organelle (nuclei, mitochondria, and endoplasmic reticulum) segmentation tasks in terms of quality metric values with NRMSE, SSIM, PCC, DICE, and mIOU. Compared with the Unet and UwUnet prediction methods, our proposed MPFnet method outperformed all quality metric values with NRMSE, SSIM, PCC, DICE, and mean IOU. Especially for the nuclei prediction task, MPFnet achieved a 3.2% improvement over the UwUnet method and a 22.7% enhancement over the Unet method in terms of mIOU. For the mitochondria prediction task, MPFnet achieved a 2.6% improvement over the UwUnet method and a 19.8% enhancement over the Unet method in terms of mIOU. To sum up, through the comprehensive analysis of mIOU quantitative indicators corresponding to different methods (Table 1, Table 2 and Table 3), we can draw a more accurate conclusion from the quantitative standard that our method is the best of all tested methods.

Furthermore, not only was the mIOU metric used as an evaluation index, but the NMSE quantitative indicators were also utilized to compare and analyze the performance of different prediction models in this section. For the nuclei prediction test set shown in Table 1, Table 2 and Table 3, the obtained NMSE of the MPFnet model was 0.192, which has less than half that of the classical UNet model, and a 4.48% reduction compared to the UwUnet model. On the mitochondria prediction test set, the lowest NMSE value was acquired by the MPFnet model as 0.206, achieving a 59.69% depression compared to the classical UNet model and a 5.07% decrement compared to the UwUnet model. As for the endoplasmic reticulum prediction test set, the obtained NMSE of the MPFnet model was also lowest at 0.187, which has less than half that of the classical UNet model, and 16.89% diminution compared to the UwUnet model.

In order to further explore the prediction performance of different models, we give more calculations of the correlated pixels for the obtained organelle fluorescence images and the predicted organelle fluorescence from SRS microscopy images with three variants of deep learning-based predictors, respectively. Another quantitative parameter, PCC, was also applied to detect the consistency between the prediction results and the target values to further study the variability. From Table 1, Table 2 and Table 3, it can be observed that the MPFnet model shows top performance in terms of PCC coefficient. The predicted PCC value of the nuclear validation set was as high as 0.908 for our proposed MPFnet model. Similar results were also observed in mitochondrial samples and endoplasmic reticulum test sets (Pearson’s *r* = 0.903, 0.911, respectively). In terms of nuclei, mitochondria, and endoplasmic reticulum test sets, the PCC similarity coefficient results from MPFnet were all higher than the classical UNet performance as follows: 7.71%, 8.14%, and 6.43%, and the PCC similarity coefficient results from MPFnet were all higher than the UwUNet performance as follows: 1.79%, 2.61%, and 0.89%.

In addition, the SSIM metric was also used as the evaluation index for similarity comparisons among image prediction results. From the nuclei prediction test set shown in Table 1, Table 2 and Table 3, the obtained SSIM of the MPFnet model was 0.913, which achieved a 20.61% improvement over the classical UNet model with 0.757, and a 2.35% increment compared to the UwUnet model. Meanwhile, on the mitochondria prediction test set, we observed that the highest SSIM value was acquired by the MPFnet model as 0.915, a 20.39% improvement compared to the classical UNet model and a 3.27% increase compared to the UwUnet model. As for the endoplasmic reticulum prediction test set, the obtained NMSE of the MPFnet model was also best at 0.886, which improved 16.43% over the classical UNet model and 3.38% compared to the UwUnet model.

At the same time, another quantitative parameter, Dice, was also applied to detect the consistency between the prediction results and the target values to further study the variability. Table 1, Table 2 and Table 3 show that the MPFnet model had top performance in terms of Dice coefficients. The predicted Dice value of the nuclear validation set was as high as 0.935 for our proposed MPFnet model. Similar results were also observed in mitochondrial samples and endoplasmic reticulum test sets (Dice *r* = 0.933, 0.936, respectively). In terms of nuclei, mitochondria, and endoplasmic reticulum test sets, the Dice coefficient results from MPFnet were all higher than classical UNet performance as follows: 9.36%, 10.68%, and 9.35%, and the Dice metric coefficient results from MPFnet were all higher than the UwUNet performance as follows: 1.74%, 1.30%, and 1.19%.

## 3. Discussion

From the comparison of protein subcellular localization results among various prediction algorithms shown in Figure 10, it can be clearly observed that the Unet model was good at extracting local feature regions (red box on the second column), but experienced difficulty in capturing global representations (region of green box on the second column). Compared with CNN, the global cues were still blurred (green box on the third column), even though the local details were better for the UwUnet (red box on the third column). In contrast, the global cues were significantly enhanced (green box on the third column), and the local details were retained for the MPFnet model (red box on the third column).

To sum up, through the comprehensive analysis of all three quantitative indicators in Table 1, Table 2 and Table 3, we can draw a more accurate conclusion from the quantitative standard that our method is the best among all modules in Table 1, Table 2 and Table 3. In conclusion, our results show that deep learning creates some new opportunities for accurately predicting the location of cellular organelles from label-free cell optical images. Existing U-net-based medical image prediction methods are insufficient for catching long-range dependencies in tested images. The multiple parallel fusion predictor combines the merits of transform and UNet methods. The new multiple parallel fusion method can intelligently reveal and extract the nonlinear correlation between features to improve prediction performance. Additionally, as illustrated in Section 4.2, our deep learning approach also improves the image SNR, which offers a solution to highly suppress image artifacts and solve the distortion problems for high-speed SRS cell imaging.

## 4. Methods and Materials

### 4.1. Experiment of the Simultaneous SRS and Fluorescence Microscopy

The complete experiment and process of predicting the subcellular protein localization based on the deep learning network are shown in Figure 11. A dual-output, 80 MHz femtosecond pulsed laser (InSight, Spectra-Physics, USA) provides the pump beam (tunable from 680 to 1300 nm) and the Stokes beam (fixed at 1040 nm) for the SRS system. Both the synchronized pump beam with 798 nm and Stokes beam with 1040 nm through the time delay stage were combined on a dichroic mirror before being directed through the microscope by the two-dimensional galvo scanning. The SRS loss signal from the lock-in amplifier and fluorescence signal from the photomultiplier tube were collected simultaneously. Images were acquired with 512 × 512 pixels, and a pixel dwell time of 8 μs at each of the ten vibrational transitions.

The deep learning-based computer-aided method for detecting subcellular protein localization using a Stimulated Raman Scattering microscopic images framework consists of the following stages: Firstly, the cell sample is prepared. Later, The SRS signal and fluorescence signal of different lung cancer cell samples were collected simultaneously using Stimulated Raman Scattering microscopy. Finally, the subcellular protein localization of lung cancer cells is performed using different machine learning techniques.

Specifically, lung cancer cells (A549, from ATCC, USA) were first cultured in ATCC F-12K medium (ATCC, USA). Then, the cells were fixed using 2% paraformaldehyde after being dyed. For the prepared live cells, after installing the living cell samples, the prepared cells were imaged with Stimulated Raman Scattering microscopy. After that, fluorescence images of nuclei, mitochondria, and endoplasmic reticulum were detected with fluorescent dyes, including Hoescht 33342, MitoTracker Red CMXRos, and ER-Tracker Green of different colors. In this work, a Unet algorithm was utilized to significantly improve the signal-to-noise ratio of nonlinear optical images. The Unet architecture consists mainly of convolutional layers that take an input feature map and convolves it with a set of filters to produce an output feature map. Based on the minimum loss from the optimizer on the training set, we chose the best hyperparameters and denoised the cell imaging test set. After denoising and enhancing the collected image, the 1200 processed cell sampling images were divided into two subsets, 80% of which was used for training, and the deep learning algorithm based on different algorithms was used to train the model. The remaining 20% of the subset was used as a test set to validate the model.

### 4.2. Protein Subcellular Localization Based on Multiple Parallel Fusion Deep Networks

The bottleneck in predicting subcellular protein locations of SRS cell imaging lies in modeling complicated relationships concealed beneath the original cell imaging data owing to the spectral overlap information from different protein molecules. Concerning the above issue, a multiple parallel fusion (MPF) Deep Network for Protein Subcellular Localization from Label-free live cell imaging is proposed to overcome the crowded and highly convoluted information, as shown in Figure 12. The main processes are as follows:

Step 1. According to the lung cancer cell imaging experiment, establish and store a lung cancer cells SRS imaging data set.

Step 2. Preprocess lung cancer cell SRS raw data sequences for the deep-learning- enabled image denoising and restoration [50,51,52].

Step 3. Build an integrated multiple parallel fusion deep networks framework.

Step 3.1. Construct independent transformer and CNN fusion models corresponding to different protein subcellular sites and fluorescence imaging labels.

Step 3.2. Train multiple parallel fusion models. Evaluate the protein subcellular location prediction performance according to the quantified metrics.

Step 4. Apply the different cell data sets to optimize the model parameters and find the optimal model combination.

Step 5. Locate the subcellular protein sites by using new cell data.

### 4.3. Multiple Parallel Fusion Neural Network Architecture and Implementation

As shown in Figure 13, the multiple parallel fusion network consists of mainly four components, which include two parallel branches and one multiple parallel fusion model in order to process input cell imaging information differently: (1) CNN branch, which gradually increases the receptive field and encodes the feature from local to global; (2) transformer branch, which starts with global self-attention and recovers the local details at the end; (3) multiple parallel fusion module, where fused features of the same resolution are extracted from each branch; and (4) gated skip-connection, where it combined the multi-level fused feature maps and generates prediction results. Each of the components are introduced in the following sections.

#### 4.3.1. CNN Branch

As shown in Figure 13, the CNN branch adopts hierarchical feature extraction, in which the resolution of the feature map decreases with the deepening of the network while the channel number increases. We divided the whole CNN branch into four stages. Each stage is composed of multiple residual convolution blocks. Each residual convolution block contains multiple convolution layers, specification layers, and activation layers. In the first stage, we increased the number of channels of the image without reducing the resolution of the image. The next three stages are feature extraction, resolution reduction to one-half of the previous layer, and increasing the number of channels to enrich the feature information.

In experiments, CNN branches were used to obtain shallow features and retain richer original information by increasing the number of channels. In order to further obtain the global context information, the transformer branch was also employed for further accurate detection and location prediction while retaining the original information. Further, the characteristic maps of each level of the CNN branch and the characteristic map of the characteristic pyramid structure generated by the transformer are simultaneously input into the MPFnet model, which includes a CBAM block, channel attention block, spatial attention block, and convolution block for multi-feature information fusion.

It is worth noting that the CNN branch is used in the process of down sampling and combined into the MPFnet model for multi-feature information fusion with transformer output to enrich the original details. Moreover, in the up-sampling process, the high-resolution characteristic images at all levels of the encoder CNN are jump connected and added to the decoder path to obtain more detailed information and more accurate location.

#### 4.3.2. Transformer Branch

Because of CNN’s lack of translation invariance and global representation capture, and transformers good long-distance dependence capturing between sequences, we added transformer branches to the encoder. As shown in Figure 13, the transformer branch also follows the encoder-decoder structure in general. In the encoder part, the input image $x\in R^{H\times W\times C}$ with C channels is directly divided into a grid of $N=\frac{H}{M}\times \frac{W}{M}$ patches. Then, each patch is mapped into a one-dimensional vector through linear mapping $f:\;p\to e\in R^{C}$, and the H × W dimensions are flattened to form the input feature map to a linear embedding layer with output dimension *S*_0_, obtaining the raw embedding sequence $e\in R^{N\times S_{0}}$.
(15)$$X_{0}=\left[E_{x_{1}},E_{x_{1}},\ldots ,E_{x_{N}}\right]$$

To capture positional information, learnable position coding pos = [ps_1_, …, ps_N_] ∈ $R^{N\times S_{0}}$ is added at the same time to the sequence of patches to obtain the resulting input sequence of tokens.
(16)$$z_{0}=x_{0}+pos$$

The encoder in the transformer module with *L* = 8 transformer layer modules is applied to the sequence of tokens *z_0_* to generate a sequence of contextualized encoding $z_{L}\in R^{N\times S_{0}}$, and each transformer layer module includes multi-head self-attention (MSA), multi-layer perceptron (MLP) block, layer Norm block, Gelu layer, and dropout layer. Among them, the self-attention layer is the core module, which converts picture pixels into timing sequences in order to capture long-term dependence and retain global information. In the self-attention block, it first maps each input sequence $z_{L}\in R^{N\times S_{0}}$ to Q, K, and V, and then calculates the correlation between Q and K as the weight of the subsequent corresponding V.
(17)$$q=z_{l}Wq,\;k=z_{l}Wk,v=z_{l}Wv\;\;W_{q},W_{k},\;W_{v}\in R^{S\times d_{}}$$
where W_Q_, W_K_, W_V_ ∈ *R^S^*^×*d*^ are the learnable parameters of three linear projection layers, and *d* is the dimension of (query, key, value).

Self-attention (*SA*) is then formulated as:(18)$$SA\left(z_{i}\right)=softmax\left(\frac{q_{i}k^{T}}{\sqrt{D_{h}}}\right)v$$
(19)$$SA\left(Z_{l-1}\right)=Z_{l-1}+\mathrm{softmax}\left(\frac{Z_{l-1}W_{Q}{\left(ZW_{K}\right)}^{⊤}}{\sqrt{d}}\right)\left(Z_{l-1}W_{V}\right).$$

*MSA* is an extension with *m* independent *SA* operations and projects their concatenated outputs:(20)$$MSA\left(Z_{l-1}\right)=\left[{SA}_{1}\left(Z_{l-1}\right);{SA}_{2}\left(Z_{l-1}\right);\cdots ;{SA}_{m}\left(Z_{l-1}\right)\right]W_{O}$$
where W_O_ ∈ R ^md×C^. *l* is typically set to C/m.

The output of *MSA* is then transformed by an *MLP* block with a residual skip as the layer output as:(21)Zl=MSALNZl−1+MLPLNMSAZl−1∈ℝL×C
where {*Z*_1_, *Z*_2_, …, *Z_L_*} is the features of transformer layers, *LN*(.) is the layer norm applied before *MSA* and *MLP* blocks.

The contextualized encoding sequence *Z_l_* contains rich semantic information used by the Decoder. For the decoder part of this work, the three steps of the progressive upsampling method are used. In the first step, the encoder feature *Z* is reshaped back from a 2D shape of $\frac{H\times W}{16\times 16}\times C_{0}$ to a 2D feature map of $s^{0}\in R^{\frac{H}{8}\times \frac{W}{8}\times C_{0}}$. In the second step, the spatial resolution is more recovered from the 2D feature map of $s^{0}\in R^{\frac{H}{8}\times \frac{W}{8}\times C_{0}}$ to $s^{1}\in R^{\frac{H}{8}\times \frac{W}{8}\times D_{1}}$. In the third step, the upsampling-convolution layer is consecutively utilized to reshaped back from $s^{1}\in R^{\frac{H}{8}\times \frac{W}{8}\times D_{1}}$ to $s^{2}\in R^{\frac{H}{8}\times \frac{W}{8}\times D_{2}}$. At last, all feature maps with different scales will be sent to multiple parallel fusion modules for coupling with the output feature of the CNN branch, respectively.

#### 4.3.3. Multiple Parallel Fusion Module

Considering the feature misalignment between CNN and transformer features, the multiple parallel fusion model (MPF) is designed as the bridge, which is applied to effectively combine the encoded features from different branches (Figure 14). Since CNN and transformer branches tend to capture features of different levels ranging from the local to global scale, MPF modules are inserted into every block to consecutively eliminate the semantic divergence between them in an interactive fashion. We take the outputs from the fourth ($h^{0}\in R^{\frac{H}{16}\times \frac{W}{16}\times C_{0}}$), third ($h^{1}\in R^{\frac{H}{8}\times \frac{W}{8}\times C_{1}}$), and second ($h^{2}\in R^{\frac{H}{4}\times \frac{W}{4}\times C_{1}}$) blocks to fuse with the results from transformer. In sum, the fused feature representation ***f****^i^*, *i* = 0, 1, 2 can be obtained by:(22)$${\hat{s}}^{i}=ChannelAttn\;\left(s^{i}\right)$$
(23)$${\hat{h}}^{i}=SpatialAttn\left(h^{i}\right)$$
(24)$${\hat{b}}^{i}=Conv\left(h^{i}W_{1}^{i}⊙h^{i}W_{2}^{i}\right)$$
(25)$$f^{i}=Residual\left(\left[{\hat{b}}^{i},{\hat{s}}^{i},{\hat{h}}^{i}\right]\right)$$
(26)$${\hat{f}}^{i+1}=Conv\left(\left[Up\left(f^{i}\right),AG\left(f^{i+1}\right)\right]\right)\;for\;i=0,1,\ldots$$
where $W_{1}^{i}\in R^{D_{i}\times L_{i}}$, $W_{2}^{i}\in R^{C_{i}\times L_{i}}$, | ⊙|is the element-wise dot product and *Conv* is a 3 × 3 convolution layer.

In the Multiple Parallel Fusion Model, the high-level CNN features and low-level transformer features were further fused with different attention mechanisms, including spatial attention block, channel-wise attention block, and residual block, as discussed in detail below.

#### 4.3.4. Spatial Attention Block

In cell images, the information importance of different positions of the subcellular image is also different. For example, the edge position information of subcellular is generally more important than that from other positions. Consequently, spatial attention is imperative to strengthen such important information. Compared with channel-wise attention, spatial attention pays more attention to the content information in the spatial position. Therefore, the goals of spatial attention lie in uncovering lateral areas of cell images, which provide the greatest contribution to the final high accuracy subcellular prediction and assign these areas higher weights. By distributing the weight in each spatial position, we find which spatial position information is most important and consequently enhance the characteristics of that part of the position, meanwhile inhibiting the extraction of noise features.

The scheme of the spatial attention module is shown in Figure 15, which is applied to identify valuable cell regions for subcellular recognition. Firstly, the original feature maps of $F_{}^{\prime }\in R^{C\times H\times W}$ are aggregated by a pair of average pooling and maximum pooling operations, which are formulated as:(27)$$F_{avg}(h,w)=\frac{1}{c_{v}}\overset{c_{v}}{\underset{c=1}{\sum }}F(c,h,w),h=1,2,\ldots ,h_{v};w=1,2,\ldots ,w_{v}$$
(28)$$F_{max}\left(h,w\right)=\max(c,h,w),h=1,2,\ldots ,h_{v};w=1,2,\ldots ,w_{v}$$

Each outputs a weight matrix of all spatial positions. Then, $F_{avg}(h,w)$ and $F_{max}\left(h,w\right)$ are concatenated and convoluted by the pyramid kernels with size 3 × 3, 3 × 3, 3 × 3, respectively, to generate the 2D spatial attention map after the sigmoid active function. The process can be defined as:(29)$$F_{cat}=f^{\left(2\times k-1\right)\times \left(2\times k-1\right)}(\left[F_{avg}^{s};F_{max}^{s}\right])$$
(30)$$F_{spa}=\overset{3}{\underset{k=1}{\sum }}\sigma (f^{\left(2\times k-1\right)\times \left(2\times k-1\right)}(\left[F_{avg}^{s};F_{max}^{s}\right]))$$
where $f^{\left(2\times k-1\right)\times \left(2\times k-1\right)}\left(.\right)$ is a convolution operation with the filter size of $\left(2\times k-1\right)\times \left(2\times k-1\right)$.

The elements of $F_{spa}$ represent the importance of the corresponding regions of the spatial domain. Subsequently, the weighted feature map *F″* is obtained by multiplying the feature map *F′* with the spatial attention map:(31)$$F^{\prime \prime }=F_{spa}⊗F^{\prime }$$
where *F″* denotes the final output 3D feature tensor of the convolutional attention module.

#### 4.3.5. Channel-Wise Attention

Note that the variety of texture information in the feature map of cell imaging still requires preliminarily removing useless redundant information to keep important texture features before weight calculation. Hence, a channel-wise attention squeeze-excitation module is introduced in this section to amend the texture and global context features from the input feature maps and give them higher weights. A diagram illustrating the structure of a squeeze-excitation block is shown in Figure 16.

Given input sample *V* to a channel-wise attention block, a set of learned convolutional filters are applied on *V* to produce corresponding feature matrix responses *U* ∈ *R*^*H*×*W*×*C*^, where *H* × *W* is the spatial dimension, and *C* is the channel dimension. We take *F_tr_*: *V* → *U*, *V* ∈ *R*^*H*′ × *W*′ × *C*′^, *U* ∈ *R*^*H* × *W* × *C*^ to be a convolutional operator.
(32)$$U=F_{tr}\left(V\right)\;$$
(33)$$u_{c}=k_{c}\;*\;V=\overset{C^{\prime }}{\underset{l=1}{\sum }}K_{C}^{l}*V^{l}$$
where ∗ denotes convolution, $k_{c}=\left[k_{c}^{1},k_{c}^{2},\ldots ,k_{c}^{C^{\prime }}\right]$ and $X=\left[x_{}^{1},x_{}^{2},\ldots ,x_{}^{C^{\prime }}\right]$ and $u_{c}\in R^{H\times W_{}}$, while $\;k_{c}^{l}$ is a 2D spatial kernel and therefore represents a single channel of *k*_c_, which acts on the corresponding channel of *V*.

Then, a squeeze operation and an excitation operation are applied on U sequentially to re-weight channel-wise feature responses. In order to squeeze the global spatial information into a channel descriptor, channel-wise global average pooling was utilized to squeeze global spatial information into a channel descriptor. Formally, a channel-wise statistic $z\in R^{C_{}}$ is generated by shrinking U through spatial dimensions *H* × *W*, where the *c*-th element of *z* is calculated by:(34)$$Z_{C}=F_{sq}\left(u_{c}\right)=\frac{1}{H\times W}\overset{H}{\underset{i=1}{\sum }}\overset{W}{\underset{j=1}{\sum }}u_{c}\left(i,j\right)$$

The channel-wise output $Z_{C}$ of the information aggregated in the squeeze operation is next used to modulate nonlinear inter-dependencies of all channels through an excitation operation. Here, a gating mechanism with a sigmoid activation is employed as follows:(35)$$s=F_{ex}\left(z,W\right)=\sigma \left(g\left(z,w\right)\right)=\sigma \left(W_{2}\delta \left(W_{1}z\right)\right)$$
where *σ* is the sigmoid function, *m* is a scaling parameter. *δ* refers to the ReLU function, $W_{1}\in R^{\frac{C}{m\;}\times C}$ and $W_{2}\in R^{C\times \frac{C}{m}}$.

The final output of the block is obtained by rescaling the transformation output U with the activations:(36)$$\tilde{V_{c}}=F_{scale}\left(u_{c},s_{c}\right)=s_{c}\cdot u_{c}$$
where ${\overset{ˇ}{V}}_{c}=\left[v_{1}^{},v_{2}^{},\ldots ,v_{C}^{}\right]$ and $F_{scale}\left(u_{c},s_{c}\right)$ refers to channel-wise multiplication between the feature map $u_{c}\in R^{H\times W_{}}$ and the scalar *s*_*c*_.

#### 4.3.6. Attention Gate Block

In the semantic segmentation for cell imaging based on transformation with CNN networks, the depth of the convolution layer is usually increased to expand the acceptance domain and capture more semantic context information. However, at the same time, it is still difficult to reduce false positive predictions, especially for small objects with large shape changes. In order to avoid this dilemma, the attention gate (AG) block is incorporated in the MPFnet, as shown in Figure 12, to highlight the significant information of the down-sampling output. The AG block uses contextual information from the gating signal $g\in R^{F_{g}}$ to prune the skip connection $s_{i}^{l}$, highlighting ROIs and therefore reducing false positive predictions. Through this module, we retain the spatial positioning information of some important objects as much as possible while maintaining a large receptive field.

The structure of an attention gate is shown in Figure 17. This AG receives two inputs, the gating signal $g\in R^{F_{g}}$ and the associated skip connection $s_{i}^{l}$ generated at that level. The gating vector signal *g*_i_ is used for each pixel *i* to determine the regions of focus that originate from the deepest layer of the neural network, where feature representation is the greatest at the cost of significant down-sampling. The gating vector contains contextual information to reduce lower-level feature responses.

For the AG gate block in Figure 17, the gating signal and skip connection $s_{i}^{l}$ are first resized and then combined to form attention coefficients $\alpha^{l}$ calculated by:(37)$$q_{\mathrm{att},\;i}^{l}=\psi^{T}\left(\sigma_{1}\left(W_{x}^{T}s_{i}^{l}+W_{g}^{T}g+b_{\mathrm{xg}}\right)\right)+b_{\psi }$$
(38)$$\alpha^{l}=\sigma_{2}\left(q_{\mathrm{att}}^{l}\left(s^{l},g;\Theta_{att}\right)\right)$$

Then, the output  ${\hat{s}}^{l}$ acquired with the element-wise multiplication between the original skip connection $s^{l}$ and the attention coefficients $\alpha^{l}$ provides spatial context highlighting ROIs, whose specific formula is as follows:(39)$${\hat{s}}^{l}=s^{l}\cdot \alpha^{l}$$

Attention coefficient, *α_i_* ∈ [0, 1] emphasizes salient image regions and significant features to preserve only relevant activations specific to the actual task.

### 4.4. Dataset

We employed a subset of SRS images in the fixed lung cancer cells (A549, from ATCC) dataset as one of our pre-trained data sources. These data sets were acquired simultaneously using image software by collecting the SRS signals from lock-in amplifiers and fluorescence signals from photomultiplier tubes [36]. For the fluorescence signals, all dyeing schemes are based on the standards provided that three different color fluorescent dyes were used to label and track the nucleus, mitochondria, and endoplasmic reticulum, respectively. The optical cell images with 512 × 512 pixels were obtained at a dwell time of 4 μs.

Another trained source of data we employed is the dataset cell images, which are acquired using GE’s IN Cell Analyzer systems [53]. These datasets were applied to test different deep learning methods in the work and evaluate their performance.

## 5. Conclusions

In this work, we have presented a methodology based on the MPF network, a novel multiple parallel fusion deep learning model for protein subcellular location prediction from SRS images. The MPFnet includes four modules: CNN branch, receptive field block, Multiple Parallel Fusion Module, transformer branch and Attention gate module. The multiple parallel fusion module is composed of three core components: spatial attention block, channel-wise attention block, and residual block. The fused multi-level encoded features from the CNN branch focus on local information, and the Transformer branch focuses on global information to eliminate the semantic divergence between them. The performance of the proposed multiple parallel fusion method was estimated and compared with other deep learning models such as UwUnet and Unet. It is shown from the experimental results that the new multiple parallel fusion deep network achieves top prediction performance and is faster compared with other deep learning methods while reducing the number of parameters, suggesting that it has great potential in the subcellular prediction of label-free cell optical images. In future work, we will further develop more advanced fusion methods to hybridize the Transform and Unet methods to further improve the performance of protein subcellular location in cell optical imaging.

## Figures and Tables

**Figure 1 ijms-23-10827-f001:**
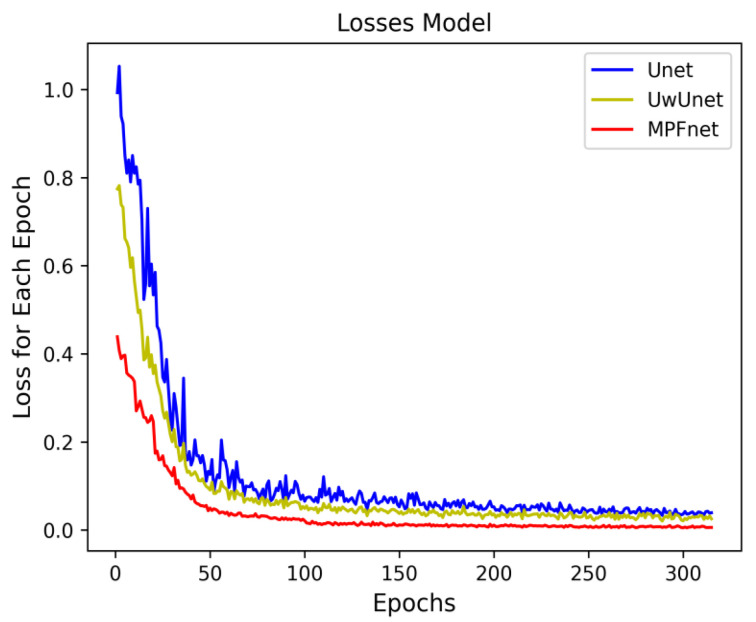
The neural network training curves for three prediction models include Unet, UwUnet, and MPFnet.

**Figure 2 ijms-23-10827-f002:**
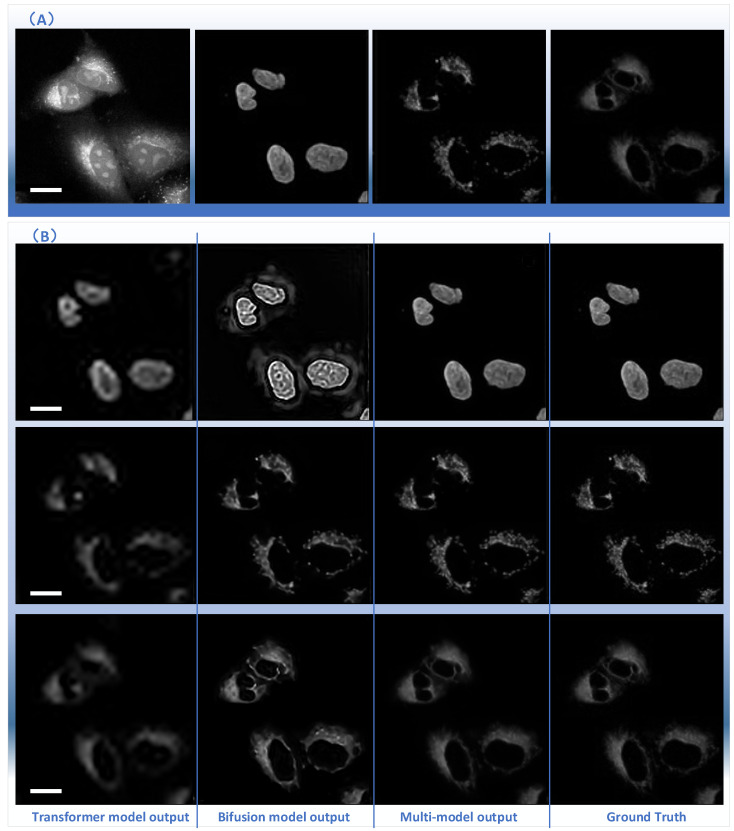
Subcellular localization prediction results of differentiation model from label-free cell imaging experiments. The results for different organelle locations, including nuclei (second column), mitochondria (third column), and endoplasmic reticulum (right) from the single raw SRS imaging cell (left), are shown in (**A**). The prediction results for organelle locations, including nuclei (top row), mitochondria (third column), and endoplasmic reticulum (bottom row), from the raw SRS imaging cells (left) are shown in (**B**). Scale bar, 25 μm.

**Figure 3 ijms-23-10827-f003:**
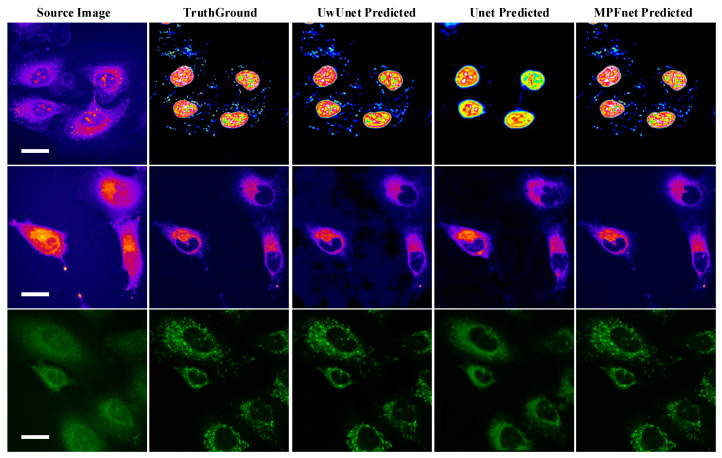
Predicted organelle fluorescence from hyperspectral SRS microscopy images by using UwUNet, U-Net, and MPFNet methods. The first column shows the Input SRS image, the second column shows the ground-truth fluorescence image, and the following three columns display the predicted fluorescence results by UwUNet, U-Net, and MPFNet, respectively, for nuclei (**top**), mitochondria (**middle**), and endoplasmic reticulum (**bottom**). Scale bar, 25 μm.

**Figure 4 ijms-23-10827-f004:**
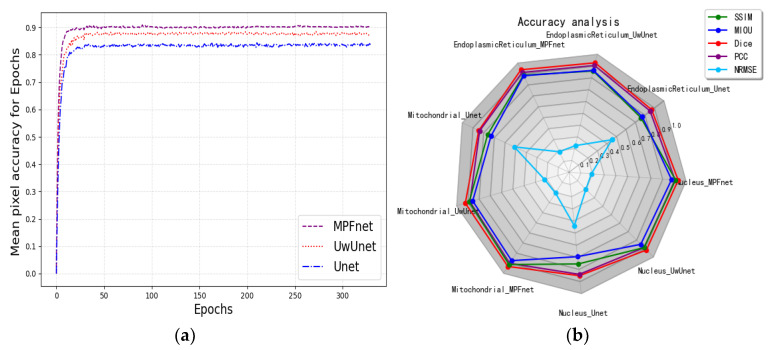
(**a**) Mean pixel accuracy comparison among different methods. Purple dash line, red dot line, and blue dash-dot line represent MPFnet, UwUnet, and Unet models, respectively. (**b**) Radar chart for quantitative comparisons of different predictive models.

**Figure 5 ijms-23-10827-f005:**
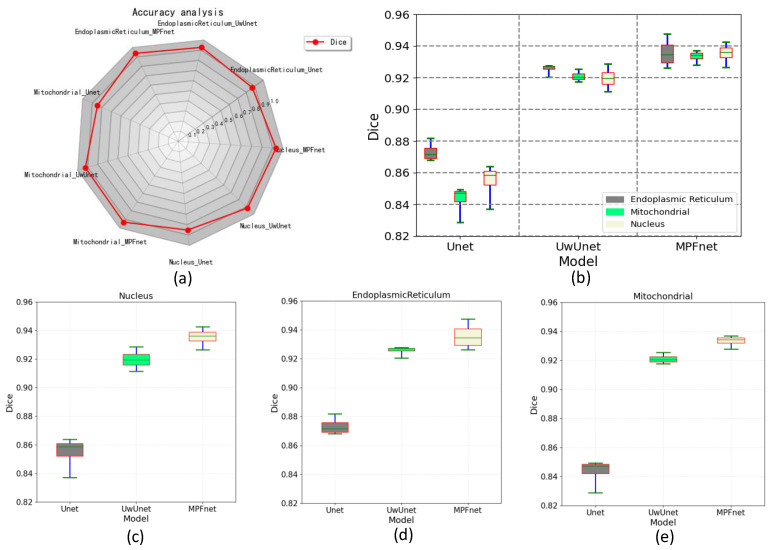
Comparing dice performance among various prediction algorithms. (**a**) Dice similarity coefficient radar chart for quantitative comparisons of different predictive models. (**b**) Box plot of dice values for all organelles (nuclei, mitochondria, and endoplasmic reticulum) prediction task set with MPFnet-based learning model and compared with that of varied deep neural network-based prediction models, such as Unet and UwUnet learning model on all. (**c**) Box plot of dice coefficient value on nuclei prediction task set with MPFnet-based learning model and compared with that of varied deep neural network-based prediction models, such as Unet and UwUnet learning model. (**d**) Box plot of dice coefficient value on endoplasmic reticulum prediction task set with MPFnet-based learning model and compared with that of varied deep neural network-based prediction models, such as Unet and UwUnet learning model. (**e**) Box plot of dice values on mitochondria prediction task set with MPFnet-based learning model and compared with that of varied deep neural network-based prediction models, such as Unet and UwUnet learning model.

**Figure 6 ijms-23-10827-f006:**
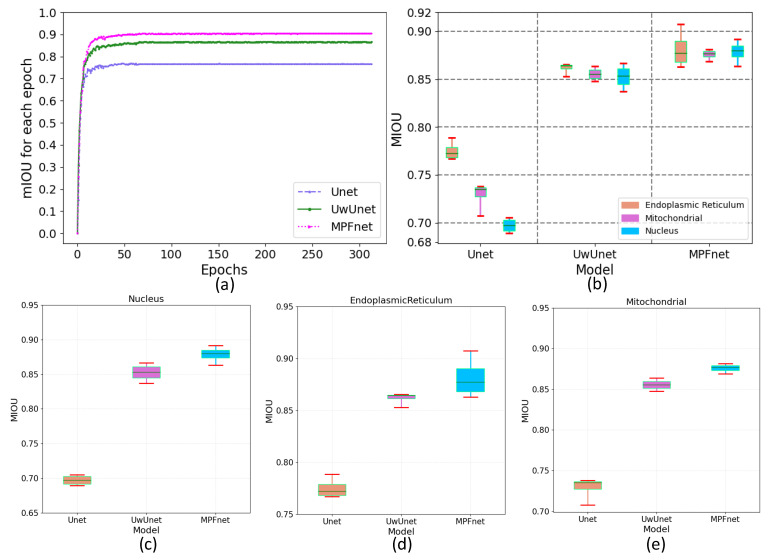
The mean intersection over union (mIOU) of different deep learning models over all SRS microscopy images in fixed lung cancer cells (A549, from ATCC) detection dataset. (**a**) mIOU for each epoch comparison among different methods. Unet (Purple triangle dash line), UwUnet (Green circle solid line), MPFnet (Pink arrow dotted line) represent Unet, UwUnet, MPFnet, respectively. (**b**) Box plot of mIOU accuracy of all organelles (nuclei, mitochondria, and endoplasmic reticulum) prediction task set with MPFnet-based learning model and compared with that of varied deep neural network-based prediction models, such as Unet and UwUnet learning models. (**c**) Box plot of mIOU accuracy on nuclei prediction task set with MPFnet-based learning model and compared with that of varied deep neural network-based prediction models such as Unet and UwUnet learning model. (**d**) Box plot of mIOU accuracy on endoplasmic reticulum prediction task set with MPFnet-based learning model and compared with that of varied deep neural network-based prediction models such as Unet and UwUnet learning model on all. (**e**) Box plot of mIOU accuracy on mitochondria prediction task set with MPFnet-based learning model and compared with that of varied deep neural network-based prediction models such as Unet and UwUnet learning models.

**Figure 7 ijms-23-10827-f007:**
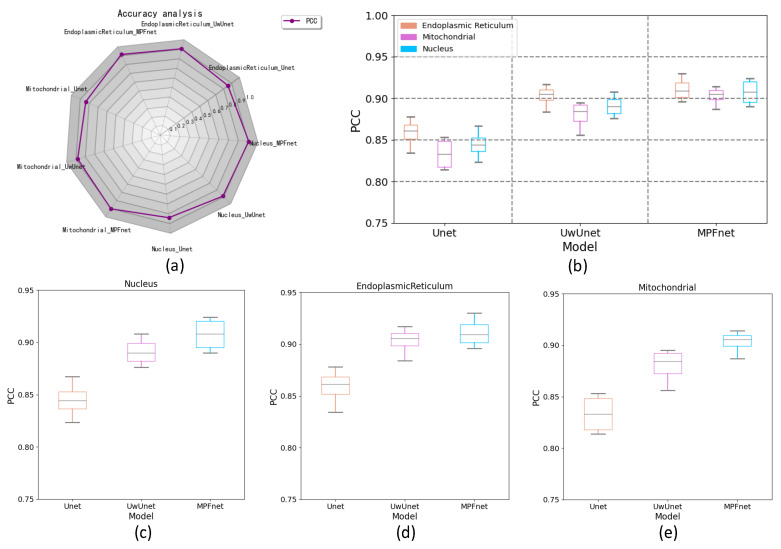
Comparing PCC performance among various prediction algorithms. (**a**) PCC radar chart for quantitative comparisons of different predictive models. (**b**) Box plot of PCC value over all organelle (nuclei, mitochondria, and endoplasmic reticulum) prediction task sets with MPFnet-based learning model and compared with that of varied deep neural network-based prediction models, such as Unet and UwUnet learning model. (**c**) Box plot of PCC value on nuclei prediction task set with MPFnet-based learning model and compared with that of varied deep neural network-based prediction models such as Unet and UwUnet learning model. (**d**) Box plot of PCC values on endoplasmic reticulum prediction task set with MPFnet-based learning model and compared with that of varied deep neural network-based prediction models such as Unet and UwUnet learning models. (**e**) Box plot of PCC values on mitochondria prediction task set with MPFnet-based learning model and compared with that of varied deep neural network-based prediction models such as Unet and UwUnet learning model.

**Figure 8 ijms-23-10827-f008:**
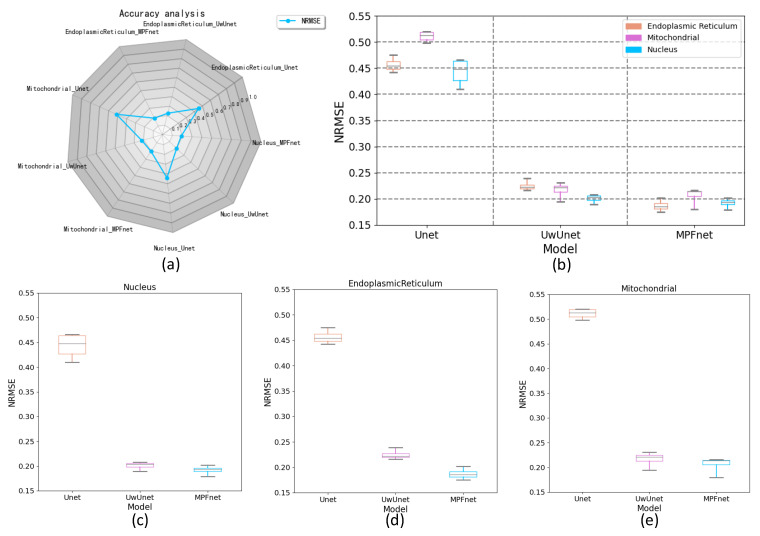
Comparing NRMSE performance among various prediction algorithms. (**a**) NRMSE radar chart for quantitative comparisons of different predictive models. (**b**) Box plot of NRMSE value over all organelles (nuclei, mitochondria, and endoplasmic reticulum) prediction task set with MPFnet-based learning model and compared with that of varied deep neural network-based prediction models such as Unet and UwUnet learning models. (**c**) Box plot of NRMSE value on nuclei prediction task set with MPFnet-based learning model and compared with that of varied deep neural network-based prediction models such as Unet and UwUnet learning model. (**d**) Box plot of NRMSE values on endoplasmic reticulum prediction task set with MPFnet-based learning model and compared with that of varied deep neural network-based prediction models such as Unet and UwUnet learning model. (**e**) Box plot of NRMSE values on mitochondria prediction task set with MPFnet-based learning model and compared with that of varied deep neural network-based prediction models such as Unet and UwUnet learning model.

**Figure 9 ijms-23-10827-f009:**
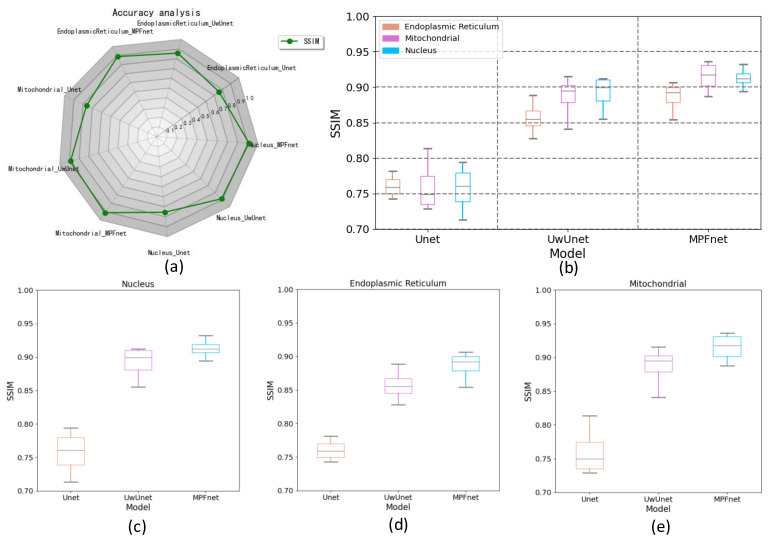
Comparing SSIM performance among various prediction algorithms. (**a**) The SSIM performance measures typical of predictive tasks. (**b**) Box plot of SSIM values in all organelle (nuclei, mitochondria, and endoplasmic reticulum) prediction task sets with the MPFnet-based learning model and compared with that of varied deep neural network-based prediction models such as Unet and UwUnet. Box plot of SSIM values on (**c**) nuclei, (**d**) endoplasmic reticulum, and (**e**) mitochondria prediction task sets with MPFnet-based learning model and compared with that of varied deep neural network-based prediction models such as Unet and UwUnet learning model.

**Figure 10 ijms-23-10827-f010:**
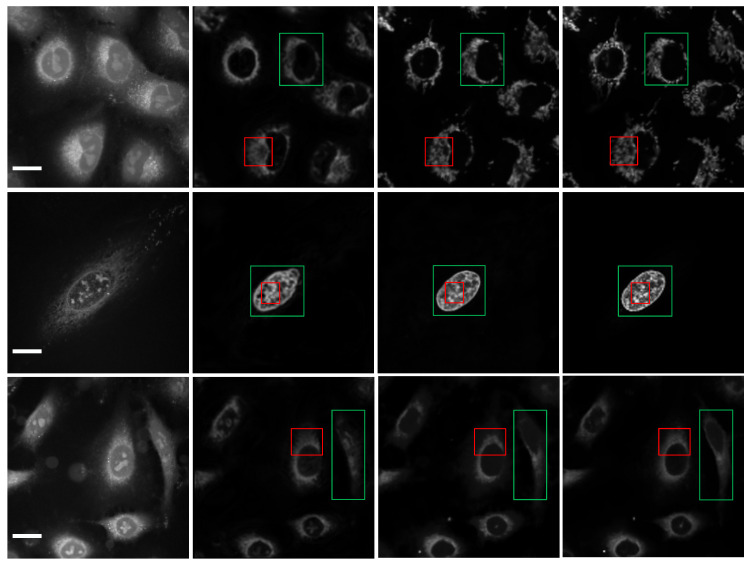
Comparison of the protein subcellular localization results from raw cell images (**first column**) among various prediction algorithms, including Unet (**second column**), UwUnet (**third column**), and MPFnent (**fourth column**). Local feature regions (red box), global representations (green box). Scale bar, 25 µm.

**Figure 11 ijms-23-10827-f011:**
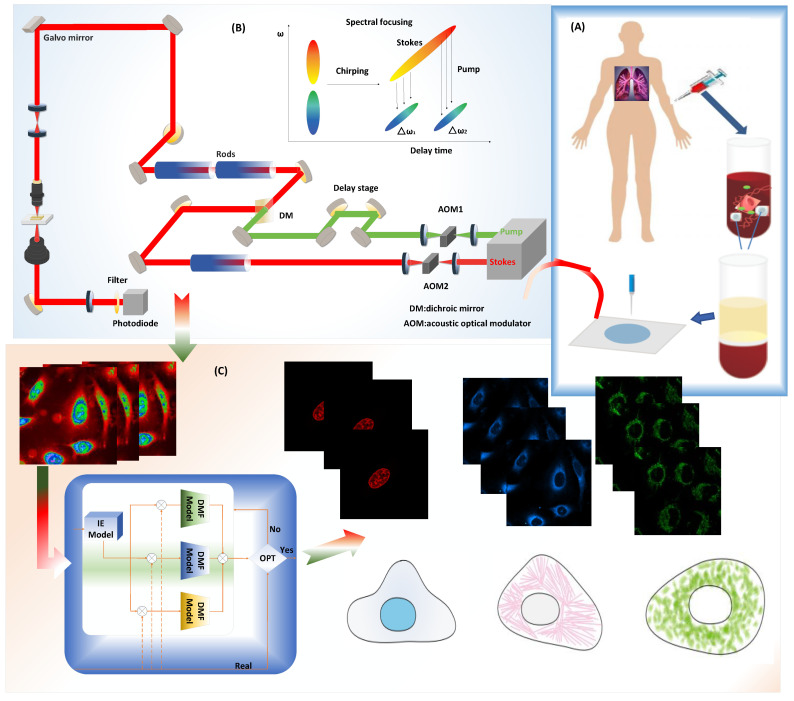
Workflow of the Single-Cell experiment by Stimulated Raman Scattering Imaging and deep learning model prediction process. (**A**) The process of prepearing the lung cancer cell sample. (**B**) Stimulated Raman Scattering microscopy setup for collecting SRS signal of the lung cancer cell samples. (**C**) Different machine learning techniques for the subcellular protein localization of lung cancer cells.

**Figure 12 ijms-23-10827-f012:**
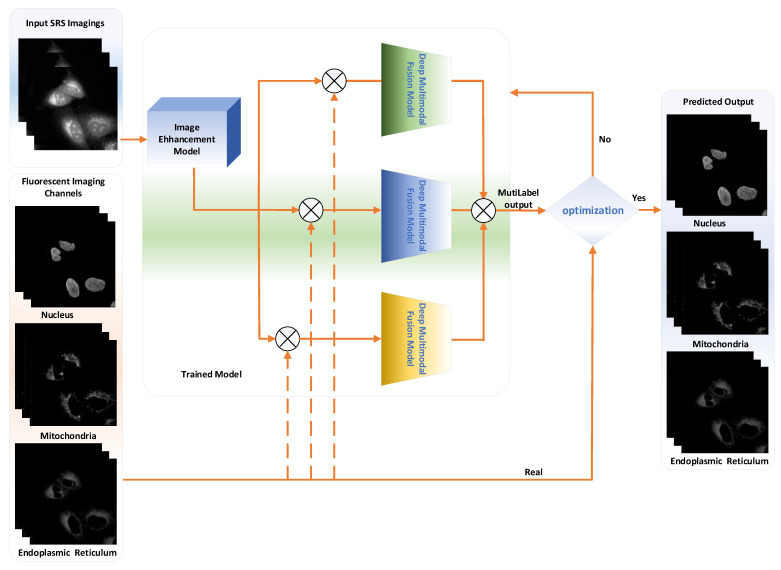
Graphical overview of our proposed deep MPF architecture. It includes five Modules: Input SRS imagings, Fluorescent Imaging Channels, Trained Model, Optimization and Predicted Output.

**Figure 13 ijms-23-10827-f013:**
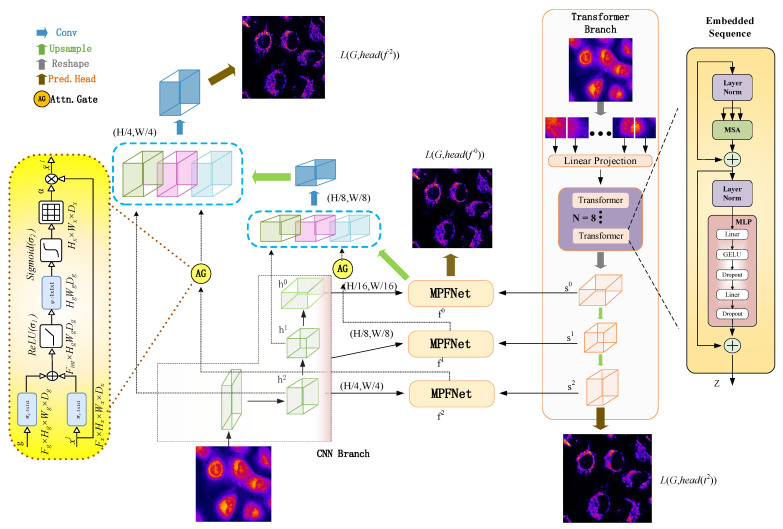
Multiple Parallel Fusion Deep Networks for the Protein Subcellular Localization from Label-free live cell imaging.

**Figure 14 ijms-23-10827-f014:**
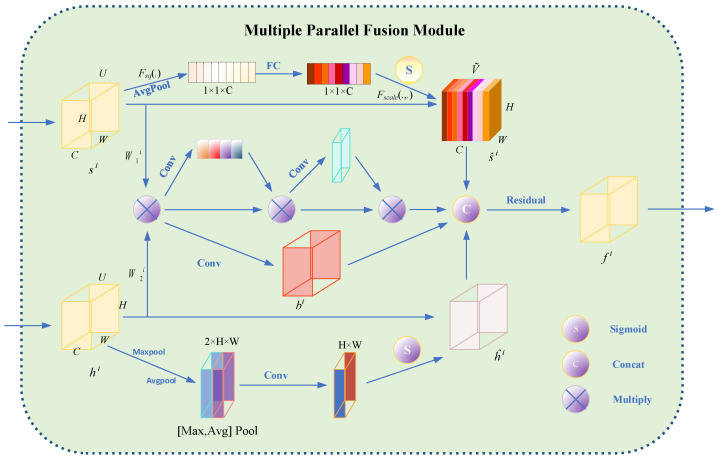
Multiple Parallel Fusion Model for the Protein Subcellular Localization from Label-free live cell imaging.

**Figure 15 ijms-23-10827-f015:**
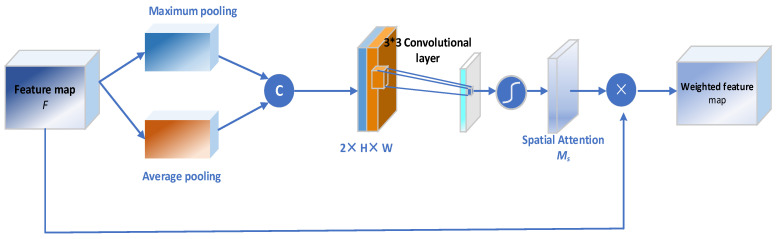
Diagram of spatial attention sub-module. As illustrated, the spatial sub-module utilizes max-pooling outputs and average-pooling outputs that are pooled along the channel axis and forwarded to a convolution layer.

**Figure 16 ijms-23-10827-f016:**
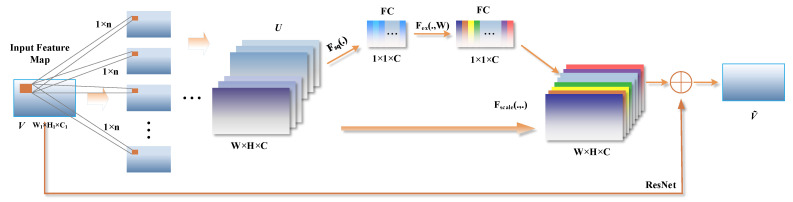
A structure scheme of the Channel-wise Attention based on squeeze-excitation block.

**Figure 17 ijms-23-10827-f017:**
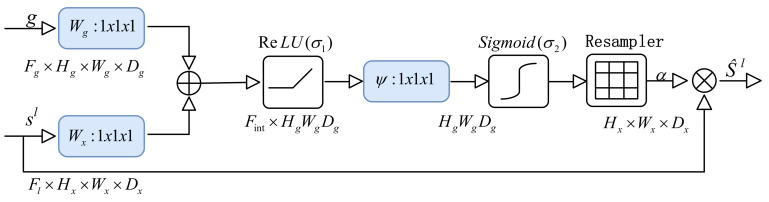
The structure of the attention gate block to highlight the significant information of the down-sampling output.

**Table 1 ijms-23-10827-t001:** Comparison of quality measures for label-free prediction results with MPFnet model. Here, ↓ indicates that the lower the index value, the better the performance; ↑ indicates that the higher the index value, the better the performance of the model.

Organelle/ Model	MFPnet Model				
	NRMSE↓	SSIM↑	PCC↑	Dice↑	mIOU↑
Nucleus	0.192 ± 0.013	0.913 ± 0.019	0.908 ± 0.018	0.935 ± 0.009	0.879 ± 0.014
Endoplasmic Reticulum	0.187 ± 0.015	0.886 ± 0.032	0.911 ± 0.019	0.936 ± 0.011	0.881 ± 0.016
Mitochondrial	0.206 ± 0.028	0.915 ± 0.028	0.903 ± 0.016	0.933 ± 0.005	0.876 ± 0.007

**Table 2 ijms-23-10827-t002:** The prediction result measures of protein subcellular localization using the UwUnet model. Here, ↓ indicates that the lower the index value, the better the performance; ↑ indicates that the higher the index value, the better the performance of the model.

Organelle\Model	UwUnet Model				
	NRMSE↓	SSIM↑	PCC↑	Dice↑	mIOU↑
Nucleus	0.201 ± 0.012	0.892 ± 0.037	0.892 ± 0.016	0.919 ± 0.009	0.852 ± 0.015
Endoplasmic Reticulum	0.225 ± 0.014	0.857 ± 0.030	0.903 ± 0.019	0.925 ± 0.005	0.861 ± 0.009
Mitochondrial	0.217 ± 0.017	0.886 ± 0.046	0.880 ± 0.024	0.921 ± 0.004	0.854 ± 0.008

**Table 3 ijms-23-10827-t003:** Comparison of quality measures for label-free prediction results with the Unet model. Here, ↓ indicates that the lower the index value, the better the performance; ↑ indicates that the higher the index value, the better the performance of the model.

Model/Organelle	Unet Model				
	NRMSE↓	SSIM↑	PCC↑	Dice↑	mIOU↑
Nucleus	0.442 ± 0.024	0.757 ± 0.044	0.843 ± 0.024	0.855 ± 0.018	0.716 ± 0.008
Endoplasmic Reticulum	0.454 ± 0.022	0.761 ± 0.021	0.856 ± 0.022	0.873 ± 0.008	0.771 ± 0.013
Mitochondrial	0.511 ± 0.013	0.760 ± 0.053	0.835 ± 0.021	0.843 ± 0.015	0.731 ± 0.021

## Data Availability

The data that support the findings of this study are available from the corresponding author upon reasonable request.

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
