# Peer review of "Multiple Parallel Fusion Network for Predicting Protein Subcellular Localization from Stimulated Raman Scattering (SRS) Microscopy Images in Living Cells"

_ijms, 2022, doi:10.3390/ijms231810827_

Round 1

Reviewer 1 Report

The goal of this manuscript is to present and characteristics of tool based on MPF network, a novel multiple parallel fusion deep learning model for protein subcellular location prediction from stimulated Raman scattering (SRS) microscopy images. The authors have done a good job at describing the problem, the methods and the results. The manuscript has high formal standard.

GENERAL COMMENTS:
TITLE
The paper title is well stated, it is informative and concise.

ABSTRACT, INTRODUCTION
Abstract and Introduction were well written, and very good presenting the subject and research problem.

MATERIAL AND METHODS
Material and research methods are presented appropriately. Experimental setup and the description in the methods section are well structured, and the all analysis is done alright.

RESULTS
The results obtained in this study are interesting. Results presented correctly.

DISCUSSION
In general, the discussion of results is correct and sufficient.

LITERATURE
The items of literature included in the paper are rather sufficient and adequate to the subject of the paper.

Author Response

We would like to express our sincere appreciation of your professional review work on our article. Those comments are all valuable and very helpful for improving our paper, as well as the important guiding significance to our researches.

Comments: The goal of this manuscript is to present and characteristics of tool based on MPF network, a novel multiple parallel fusion deep learning model for protein subcellular location prediction from stimulated Raman scattering (SRS) microscopy images. The authors have done a good job at describing the problem, the methods and the results. The manuscript has high formal standard.

Response:We deeply appreciate the reviewer’s positive evaluation of our work.Your careful review has helped to make our study clearer and more comprehensive.

Reviewer 2 Report

Wei et al. described the use of multiple parallel fusion network for predicting the subcellular localization of proteins from stimulated Raman scattering microscopy images. This work is timely and interesting and may attract the readership of International Journal of Molecular Sciences. I would recommend its acceptance for publication after the authors addressed these issues. The authors should provide more details on section 2.1., particularly the types of fluorescent dyes used, and the settings of Raman and fluorescence microscopes used. Procedures taken to denoise and enhance the collected images as well as the number of images used for training and testing the deep learning algorithm must also be described.

Author Response

We would like to express our sincere appreciation of your professional review work on our article. Those comments are all valuable and very helpful for revising and improving our paper, as well as the important guiding significance to our researches. We have studied comments carefully and have made correction which we hope meet with approval. According to your nice suggestions, we have made extensive corrections to our previous draft, the detailed corrections are listed below.

Comment 1: Wei et al. described the use of multiple parallel fusion network for predicting the subcellular localization of proteins from stimulated Raman scattering microscopy images. This work is timely and interesting and may attract the readership of International Journal of Molecular Sciences. I would recommend its acceptance for publication after the authors addressed these issues.

Response: We appreciate the reviewer’s positive evaluation of our work, and deeply appreciate the reviewer’s suggestion. Those precious comments and advice are all valuable and very helpful for revising and improving our paper.

Comment 2:The authors should provide more details on section 2.1., particularly the types of fluorescent dyes used, and the settings of Raman and fluorescence microscopes used.

Response:We deeply appreciate the reviewer’s suggestion. According to the reviewer’s comment, we provided more details about particularly the types of fluorescent dyes used, and the settings of Raman and fluorescence microscopes used on section 2.1.

Comment 3:Procedures taken to denoise and enhance the collected images as well as the number of images used for training and testing the deep learning algorithm must also be described.

Response: We are grateful for the suggestion. As suggested by the reviewer, we have added more details of Unet model taken to denoise and enhance the collected images as well as the number of images used for training and testing the deep learning algorithm (Line 122-137, page 4).

Reviewer 3 Report

Staining with different fluorescent dyes is a widespread tool for investigating cell structure but sometimes there is a need in analyzing living cells without any treatment. Surely it would be good if label-free imaging is supplemented with a tool that recognize cell structures with high sensitivity. This research is about one of these methods – multiple parallel fusion network that is more precise compared with others.

It s a well written text, however I have some comments. First of all there is chaos with spaces between words and brackets, words and points. Lines 11, 34, 39, 51, 68, 75 etc. Check the whole text. The same chaos is references. lines 831, 842, 843, 845 - there are several different styles. 

Choose one style: Figure or Fig…..Figure 1 (a) or Figure 1 a. There are many different variants.

Line 779 – it seems the talk is about Figure 17. Also give the names of methods on a picture as on figure 10.

In results MPFnet method is compared with Unet and UwUnet methods but nothing is said about these two methods in the introduction. What are the differences between them, disadvantages and etc.

MPFnet or MPF-net? -figure 10 and lines 520, 521

Lines 542-569 – there is no need in counting all the number meanings of Dice similarity for each method. It is better to say Dice similarity of nuclei prediction is the highest for MPFnet and the lowest for Unet and etc.

Line 595 – it is not “can be shown” it  is “it is shown” because there is a figure and readers see everything on it.

In the description of figures 14, 15 and 16 there is no need to give certain numbers. It is enough to say higher, lower comparing with something or similar.

Line 101 – give the country of manufacture ATCC. Also give the manufacture for medium used.
